# Binding Specificity and Oligomerization of TSWV N Protein in the Western Flower Thrips, *Frankliniella occidentalis*

**DOI:** 10.3390/v17060826

**Published:** 2025-06-07

**Authors:** Falguni Khan, Eticha Abdisa, Niayesh Shahmohammadi, Yonggyun Kim

**Affiliations:** Major in Plant Medicals, School of Life Sciences, Gyeongkuk National University, Andong 36729, Republic of Korea; falgunikhan2942@gmail.com (F.K.); abdi.eti06@gmail.com (E.A.); niayeshshahmohammadi@gmail.com (N.S.)

**Keywords:** TSWV, replication, N protein, nucleocapsid, *Frankliniella occidentalis*

## Abstract

Tomato spotted wilt virus (TSWV) is a highly destructive plant pathogen and transmitted by several thrips including the western flower thrips, *Frankliniella occidentalis*. A structural N protein encoded in the viral genome represents the nucleocapsid protein by binding to the viral RNA genome. However, it remains unknown how the RNA-binding protein specifically interacts with the viral RNA from host RNAs in the target cells. To study the molecular basis of N function, we produced the protein in *Escherichia coli* and the resulting purified recombinant protein was used to investigate the protein–RNA interactions. The recombinant N protein migrated on agarose gel to the anode in the electric field due to its high basic isoelectric point. This electrostatic property led N protein to bind to DNA as well as RNA. It also bound to both single-stranded (ssRNA) and double-stranded RNA (dsRNA). However, when the total RNA was extracted from plant tissues collected from TSWV-infected host, the RNA extract using the recombinant N protein was much richer in the TSWV genome compared to that without the protein. To investigate the specificity of N protein to ssRNA, the three-dimensional structure was predicted using the AlphaFold program and showed its trimeric oligomerization with the binding pocket for ssRNA. This was supported by the differential susceptibility of N protein with ssRNA and dsRNA against RNase attack. Furthermore, a thermal shift assay to analyze the RNA and protein interaction showed that ssRNA strongly interacted with N protein compared to dsRNA. In addition, the *N* gene was expressed along with the multiplication of the viral RNA genome segments from the segment-specific fluorescence in situ hybridization analysis in different tissues during different developmental stages of the virus-infected *F. occidentalis*. These results suggest that the functional trimeric N proteins bind to the viral RNA to form a basic nucleocapsid structure at a specific virus-replicating compartment within the host cells.

## 1. Introduction

*Orthotospovirus tomatomaculae*, tomato spotted wilt virus (TSWV), a member of the *Tospoviridae* family, is a highly destructive plant pathogen affecting a wide range of crops [1,2]. It is transmitted only by a few thrips among more than 7000 species [3]. The western flower thrips, *Frankliniella occidentalis*, is a vector of the virus in a circulative and persistent manner [4]. The virus infects the young larval stage via feeding on the infected host plants and translocates from the gut epithelium to the salivary gland during development into adults [5]. The virus in the salivary gland of the adults is then translocated to host plants via saliva during extra-oral digestion of the thrips [6].

TSWV forms a spherical and enveloped particle, and its genome consists of three single-stranded RNA segments, large (L), medium (M), and short (S), which follow a negative/ambisense coding strategy [7]. The L segment encodes the RNA-dependent RNA polymerase (RdRp), which is essential for replication and transcription [8]. The L RNA segment is in a negative sense while the M and S RNAs have an ambisense organization. The M segment encodes two glycoproteins (Gn and Gc) and a non-structural movement protein (NSm), whereas the S segment encodes the capsid (N) protein and a second non-structural protein (NSs) [9,10].

The N protein of TSWV is a critical component of the virus, playing multiple functional roles in its life cycle [11,12]. The open reading frame (ORF) of the *N* gene encodes 258 amino acids, with distinct functional motifs, including an RNA-binding domain rich in positively charged amino acids, a lysine-rich nuclear localization signal, and an SR-rich motif [9]. The N protein consists of the following two independent structural domains: the N-terminal domain, which is primarily involved in RNA binding, and the C-terminal domain, which is responsible for self-association [13]. This protein is responsible for recognizing and packaging viral RNA into ribonucleoprotein complexes, facilitating viral mRNA transcription, replication, and host immune evasion [14]. It encapsidates viral RNA, protecting it from degradation by host cellular defenses and RNA-silencing pathways [15]. The encapsidation process is necessary for the recruitment of viral RNA into replication compartments, where the RdRp can function efficiently. Additionally, the N protein interacts with Gn/Gc or NSm, influencing virus assembly and dissemination [16,17]. For example, the N protein interacts with NSm, potentially guiding viral RNA through plasmodesmata for cell-to-cell movement [16]. In addition, the N protein undergoes conformational changes upon RNA binding, which may facilitate the switch between transcription and replication [9]. This process ensures efficient amplification of the viral genome while maintaining regulatory control over viral mRNA production.

Structural analyses have shown that the N protein forms higher-order oligomers, suggesting its ability to efficiently encapsidate viral RNA via its RNA-binding domains [9]. Interestingly, the N protein does not display strict sequence specificity when binding RNA. This feature allows it to interact with both viral and host RNAs, raising questions about how the virus ensures preferential binding to its own genomic RNA [16]. Thus, the precise mechanism by which the N protein selectively binds viral RNA over host RNA in insect and plant cells remains poorly understood.

This study analyzed the binding affinity of the N protein to different nucleic acids using a recombinant form purified from a bacterial expression system. To clarify its binding specificity, its three-dimensional conformation was predicted to understand its higher-order oligomerization. The functional structure of the N protein was assessed by a thermal shift assay to determine its binding affinity. Finally, this study monitored the viral replication in different tissues of *F. occidentalis* along with the expression profile of the *N* gene using fluorescence in situ hybridization (FISH).

## 2. Materials and Methods

### 2.1. Insect and Plant Rearing

A laboratory colony of *F. occidentalis* was obtained from the Department of Crop Protection at the National Institute of Agricultural Sciences (Wonju, Republic of Korea) and reared on germinated kidney beans (*Phaseolus coccineus* L.) under controlled conditions: 25 ± 2 °C, 16:8 h (L:D), and 65 ± 5% relative humidity. Eggs laid on these beans were transferred to new breeding dishes (100 × 40 mm, SPL Life Sciences, Pocheon, Republic of Korea). Fresh germinated beans were added to the breeding dishes daily to support the development and reproduction of the thrips.

### 2.2. TSWV Infection of Thrips

TSWV was isolated from hot pepper leaves exhibiting typical viral symptoms, including ring spots, in Andong, Korea, and verified with an Immunostrip TSWV kit (Agdia, Elkhart, IN, USA). Approximately 100 mg of the plant tissues were homogenized in 1 mL of filter-sterilized (0.22 µm pore size) phosphate-buffered saline (PBS, pH 7.4) and centrifuged at 14,000× *g* for 5 min. The supernatant was employed as the virus suspension. The viral infection followed the method described by Shahmohammadi et al. [17]. Briefly, prior to the immune challenge, L1 or L2 larvae starved for 1 h. The viral infection was administered via a feeding method in which sprouted bean seed kernels were immersed in 1 mL of the virus suspension for 5 min and subsequently dried for 10 min under aseptic conditions. The infected kernels were then placed in the breeding dish where the test insects fed for 12 h and were used as viruliferous thrips.

### 2.3. Bioinformatics

The TSWV-N protein (CAD11452.1), nucleotide sequences (Z36882.1), TSWV NSs nucleotide sequences (MZ687800.1), and *F. occidentalis* glycoprotein nucleotide sequences (MH884757.1) were obtained from the NCBI GenBank (http://www.ncbi.nlm.nih.gov, accessed on 15 May 2025). Sequence alignment for TSWV-N protein was performed using ClustalW programs from Megalign software (DNAstar, Madison, WI, USA). Protein domains were predicted using EMBL-EBI (www.ebi.ac.uk, accessed on 15 May 2025) and Pfam (http://pfam.xfam.org, accessed on 15 May 2025). AlphaFold v3 (https://neurosnap.ai/, accessed on 15 May 2025) was used to construct three-dimensional structures and analyze active binding sites using the InterPro program (https://www.ebi.ac.uk/interpro/, accessed on 15 May 2025).

### 2.4. RNA Extraction, cDNA Synthesis, and RT-qPCR

Approximately 2 g of TSWV-infected hot pepper leaves were homogenized with 1 mL of Trizol (Intron Biotechnology, Sungnam, Republic of Korea) in a 1.5 mL tube (Eppendorf AG, Hamburg, Germany). Over 1000 thrips were homogenized in 500 μL of Trizol in a tube to extract insect RNA. The homogenates were centrifuged at 14,000× *g* for 5 min to obtain the supernatant, and RNA extraction was performed according to the manufacturer’s instructions. The extracted RNA was resuspended in nuclease-free water and quantified with a spectrophotometer (NanoDrop, Thermo Scientific, Wilmington, DE, USA).

For RT-qPCR of the viral genes, the extracted viral RNA was directly used for RT-PCR in a total volume of 20 μL containing 4 μL of template RNA, 2 μL of gene-specific primers, 4 μL of nuclease-free water, and 10 μL of SuPrime Script RT-PCR Premix (2×) (Genetbio, Daejeon, Republic of Korea). The reverse transcription was performed at 50 °C for 30 min. After a denaturation step at 95 °C for 5 min, the subsequent PCR proceeded according to the following temperature cycle program: 35 cycles of 95 °C for 30 s, 55 °C for 1 min, and 72 °C for 1 min. For RT-qPCR of thrips genes, cDNA was constructed using an RT Premix (Intron Biotechnology, Seoul, Republic of Korea) containing oligo-dT primer based on the manufacturer’s instruction. A PCR reaction mixture (25 µL) was then prepared for RT-PCR with 1 µL of the cDNA template (100 ng/µL), dNTPs (each 2.5 mM), 10 pmol of each forward and reverse primer, and Taq polymerase (2.5 unit/µL). Quantitative PCR (qPCR) was performed using a Real-time PCR machine (Step One Plus Real-Time PCR System, Applied Biosystems, Singapore) with Power SYBR Green PCR Master Mix (Life Technologies, Carlsbad, CA, USA). The reaction mixture (20 µL) contained 10 µL of Power SYBR Green PCR Mix, 2 µL of cDNA template (70 ng/µL), and 1 µL of each forward and reverse primer. Elongation factor-1 (*Fo-EF1*) was used as a reference gene (Appendix A). Melting curve analysis was used to evaluate the quality of the PCR results. Quantitative analysis was performed using the comparative CT (2^−∆∆CT^) method [18], where each experiment was replicated three times with individual sample preparations.

### 2.5. Multiplex PCR to Detect Viral and Insect Genes

Multiplex polymerase chain reaction analysis was used to discriminate viruliferous thrips and viral genes according to the method described by Kim et al. [19]. Thirty viruliferous thrips were homogenized in 60 μL of an RNA extraction solution (LGC Bioresearch Technologies, Hoddesdon, UK) in a 1.5 mL sample tube. Non-viruliferous thrips were used as a control. The homogenate was incubated at 95 °C for 5 min and centrifuged at 13,500× *g* for 5 min to obtain the supernatant. RNA was extracted according to the manufacturer’s instructions. PCR amplification was performed in a total volume of 20 μL containing 4 μL of template RNA, 4 μL of primers, and 8 μL of SuPrime Script RT-PCR Premix (2×) (Genetbio, Daejeon, Republic of Korea). Reverse transcription was performed at 50 °C for 30 min. After a denaturation step at 95 °C for 5 min, the subsequent PCR proceeded according to the following temperature cycle program: 35 cycles of 95 °C for 30 s, 55 °C for 30 s, and 72 °C for 1 min. Two primer sets were designed for *TSWV-N* and *Fo-Gn* to identify viral and insect genes (Appendix A). The PCR products were electrophoresed on a 2% (*w/v*) agarose gel containing 1×TAE buffer and visualized by UV light on a Gel Doc imaging system (Bio-Rad, Hercules, CA, USA). The resulting PCR product sizes were 777 bp for TSWV and 312 bp for *Fo-Gn*.

### 2.6. Protein Expression and Purification

A full-length open reading frame (ORF) of the TSWV-N gene (777 bp) was cloned into a eukaryotic expression vector pBAD/V5-His (Invitrogen, Seoul, Republic of Korea) using TA cloning. After confirming the correct orientation and sequence of the insert, the recombinant plasmid was used to transform *Escherichia coli* Top10, which was then cultured on LB agar plates supplemented with ampicillin (100 μg/mL, LB + amp).

For overexpression of the insert gene, a single loop of the recombinant bacterial colony was inoculated into 30 mL of LB + amp broth and incubated overnight at 37 °C in a shaking incubator at 170 rpm. The following day, 25 mL of each overnight culture was transferred into 1 L of fresh LB + amp broth. Cultures were grown at 37 °C with shaking (170 rpm) until the optical density at 600 nm (OD_600_) reached 0.7–0.8, which typically took 4–5 h. Protein expression was then induced by adding 2 mL of 20% arabinose solution, followed by continued incubation under the same conditions for an additional 6 h. After induction, the bacterial cells were harvested by centrifugation at 8000 rpm for 30 min at 4 °C. The supernatant was discarded, and the resulting pellet was transferred to a 50 mL Falcon tube. To lyse the cells, 20 mL of a double detergent extraction buffer (50 mM Tris-HCl, pH 8.0; 150 mM NaCl; 0.02% NaN_3_; 0.10% SDS; 1% Igepal CA-630; and 100 μg/mL PMSF) was added. The cells were then lysed by sonication on ice for 30 min. The lysate was then centrifuged at 14,000 rpm for 20 min at 4 °C to remove cell debris.

The supernatant was incubated with 2 mL of nickel nitrilotriacetic acid resin (HisPurTM Ni-NTA, Thermo Scientific, Hampton, NH, USA) at 4 °C for 2 h and then loaded onto an affinity chromatographic column (Bio-Rad). The Ni-NTA resin was washed ten times with a 10× wash buffer (1 M NaH_2_PO_4_, 29.21 g NaCl, 5 M imidazole, pH 8.0). The proteins were eluted with 10 mM, 100 mM, 200 mM, and 400 mM imidazole elution buffer (10× washing buffer, 100 mM PMSF, 5 mM imidazole, pH 8.0, and double deionized water).

### 2.7. Western Blot Analysis

Protein amounts were determined using Bradford method [20] using bovine serum albumin as a standard. Protein samples were prepared with 6× SDS loading buffer (0.3 M Tris-HCl, 0.6 M DTT, 10% SDS, 0.06% Bromophenol blue, and 50% Glycerol). After boiling for 5 min, the protein samples were separated by electrophoresis into 10% SDS-polyacrylamide gels and transferred onto a nitrocellulose membrane. The antigens on the membrane were incubated with an anti-V5 antibody against TSWV-N (1:5000 *v/v* dilution) (Sigma-Aldrich Korea, Seoul, Republic of Korea) and subsequently probed with anti-mouse IgG (γ-chain specific produced in goat) conjugated with alkaline phosphatase (1:10,000 dilution; Sigma-Aldrich Korea). Finally, the membrane was soaked in BCIP/NBT (SIGMAFAST^TM^, Sigma-Aldrich Korea) for 5–10 s to visualize the positive band.

### 2.8. RNA Preparation Using In Vitro Transcription

Viral (*TSWV-NSs*: MZ687800.1, GenBank accession number) and thrips (*Fo-Gn*: MH884757.1, GenBank accession number) genes were used to prepare RNAs. The ORF (1.4 kb) of *TSWV-NSs* was amplified by PCR to serve as a template for RNA synthesis. In contrast, a partial gene (312 bp) of *Fo-Gn* was amplified. These PCRs were performed with forward and reverse gene-specific primers containing T7 promoter sequences at their 5′ ends (Appendix A). To prepare dsRNA, PCR products were produced using T7 promoter-tagged primers for both forward and reverse strands and used as templates for T7 RNA polymerase with the MEGAscript RNAi kit (Ambion, Austin, TX, USA). In contrast, single-stranded RNAs (ssRNAs) were prepared using PCR products containing single T7 promoters on either strand, which were then used for the in vitro transcription.

### 2.9. Electrophoretic Mobility Shift Assay (EMSA)

To evaluate the binding affinity of the TSWV-N protein to nucleic acids, EMSA was conducted using different nucleic acids. For the binding assay, 10 μL of RNA or DNA (100 ng/μL) was mixed with 10 μL of TSWV-N protein (616 ng/μL) and incubated at room temperature for 1 min before being subjected to electrophoresis on a 1% agarose gel to visualize RNA–protein complexes. To assess dose-dependent binding, the same concentration of RNA or DNA (10 μL, 100 ng/μL) of *Fo-Gn* was incubated with increasing concentrations of TSWV-N protein (0, 0.1, 1, 10, 100, and 616 ng/μL) for 1 min at room temperature before electrophoresis. Additionally, EMSA was performed using dsRNA, +ssRNA, or −ssRNA of *TSWV-NSs*, where 10 μL of each nucleic acid (100 ng/μL) was mixed with 10 μL of TSWV-N protein (616 ng/μL) and incubated for 1 min at room temperature before electrophoresis.

### 2.10. Susceptibility of N Protein and RNA Complex to RNase A

The dsRNA was heated at 95 °C for 5 min and cooled on ice for 2 min to produce ssRNA. A total of 10 μL of the purified TSWV-N protein (616 ng/μL) was added to 10 μL of dsRNA or ssRNA (100 ng/μL). The mixture was added to 5 μL of RNase A (Sigma-Aldrich Korea, Seoul, Republic of Korea) and incubated on ice for 15 min. The mixture was centrifuged at 14,000 rpm for 3 min, and then electrophoresis was performed on 1% agarose gel.

### 2.11. Pull-Down of RNA with TSWV-N Protein

Total RNAs were heated at 95 °C for 5 min and cooled on ice for 2 min before being incubated with the purified TSWV-N protein. A total of 50 μL of the purified TSWV-N protein was added to 25 μL of total RNA. HisPur^TM^ Ni-NTA Resin (Thermo Scientific) was prepared by washing with 1x binding buffer which has been treated within DEPC-treated water. The mixture was added to 50 μL of Ni-NTA Resin and incubated on ice for 2 h at 4 °C with gentle shaking at 50 rpm. The mixture was centrifuged at 14,000 rpm for 3 min. Then, the total RNA was extracted from the pellet using Trizol.

### 2.12. Determination of the Oligomeric Structure of TSWV-N Protein

Size-exclusion chromatography was used to separate different quaternary structures of TSWV-N. A total of 616 μg of the recombinant N protein was loaded onto a column (10 cm diameter and 150 cm in length) packed with Sephadex G-100 (Sigma-Aldrich Korea) with PBS as the eluent at a flow rate of 1 mL per min. Each 500 μL fraction was measured for total protein amount by the Bradford method [20]. The peaks were analyzed by 10% SDS-PAGE and Coomassie staining. To denature the protein, 616 μg of the recombinant N protein was treated with 95 °C for 5 min and separated using the same size exclusion chromatography. To cross-link the oligomers, 616 μg of the recombinant N protein was incubated with 0.003% glutaraldehyde at 37 °C for 5 min and quenched by adding 2 µL of 1 M Tris-HCl (pH 7.5).

### 2.13. Thermal Shift Assay to Estimate Binding Affinity of TSWV-N Protein to RNAs

A thermal shift assay was carried out using a Thermal Shift dye kit (Applied Biosystems, Foster City, CA, USA). Briefly, the reaction mixture (20 μL) contained protein thermal shift dye (2.5 μL), protein thermal shift buffer (5 μL), TSWV-N protein (10 μL, 500 ng), and dsRNA or ssRNA (2.5 μL). A melting curve experiment was conducted using a step one real-time PCR system (Applied Biosystems). The thermal profile was obtained by heat treatment at 25 °C for 2 min and then at 99 °C for 2 min. Melting temperatures resulting from the heat treatment were plotted using SigmaPlot 10.0 (Systat Software, San Jose, CA, USA).

### 2.14. Fluorescence In Situ Hybridization (FISH)

By feeding TSWV to the L1 stage, the experiment was carried out on the larval, pupal, and adult phases. Gut tissues from thrips were dissected and placed on sterile glass slides and incubated for an hour for fixation at room temperature using 4% paraformaldehyde. After being rinsed with 1× PBS (phosphate-buffered saline), the midgut was permeabilized for one hour at room temperature using 2% Triton X-100 in PBS. Following another PBS wash, the guts were immersed in a 2× sodium saline citrate (SSC) solution and left to incubate for an hour at 42 °C with 25 μL of pre-hybridization buffer (containing 2 μL yeast tRNA, 2 μL 20× SSC, 4 μL dextran sulfate, and 2.5 μL 10% SDS (sodium dodecyl sulfate)) in a dark and humid chamber. A hybridization buffer (containing 5 μL deionized formamide and 1 μL fluorescein-labeled oligonucleotide in 19 μL pre-hybridization buffer) was then added in place of the pre-hybridization buffer. Utilizing high-performance liquid chromatography (HPLC), several DNA oligonucleotide probes that target distinct genes were purified (Bioneer, Daejeon, Republic of Korea) (Appendix A). For hybridization, the slides were then covered with RNase-free coverslips and left in a humid environment at 42 °C for approximately 16–17 h. After hybridization, the midguts were incubated for 5 min at room temperature with 4× SSC containing 1% Triton X-100 after being twice washed for 10 min each. After three rounds of washing with 4× SSC, the gut samples were incubated for 1 h in a dark environment at 37 °C with 1% anti-rabbit-FITC or Rhodamine-conjugated antibody (Thermo Scientific) in PBS. The midguts were air-dried after incubation and then rinsed twice with 4× SSC for 10 min each and once with 2× SSC. A drop of 50% glycerol was added, and the samples were allowed to sit at room temperature for 15 min before being covered with coverslips and examined at ×200 magnification using a Leica DM2500 fluorescent microscope.

### 2.15. Statistical Analysis

All experiments in this study were conducted with three individual replications. The results were plotted using SigmaPlot 10.0. Statistical analysis was performed using PROC GLM of the SAS program Version 6.03 [21] with a one-way analysis of variance (ANOVA). Significant differences among the means were determined using the LSD test at a Type I error of 0.05, indicated by different letters.

## 3. Results

### 3.1. Purification of a Recombinant Protein of TSWV-N Using a Bacterial Expression System

The amino acid sequence of TSWV-N was predicted to contain 258 amino acid residues (Figure 1a). It is rich in lysine and arginine, which constitute 14.3% of the total amino acid composition. Its secondary structure was predicted to have 13 alpha helix motifs and 2 beta pleated-sheet motifs. Notably, the RNA-binding residues (indicated by boxes in Figure 1a) are conserved along with other plant viral N proteins.

TSWV-N ORF was cloned into an expression vector and then transformed into *E. coli* (Figure 1b). After induction with L-arabinose, 10% resolving SDS-PAGE gel separated bacterial proteins, including the recombinant TSWV-N in the cell pellet fraction. Subsequent affinity chromatography using Ni-NTA agarose resin yielded a purified recombinant TSWV-N protein, which was confirmed by an immunoblot specific to the V5 tag (See the arrow in the Western blot).

### 3.2. Binding Affinity of TSWV-N Protein to Nucleic Acids

To assess the binding affinity of TSWV-N protein to nucleic acids, we incubated the Fo-Gn dsRNA with the TSWV-N protein and ran the samples on the agarose gel to determine the binding affinity (Figure 2a). The dsRNA alone was detected at about 500 bp on the 1% agarose gel. The protein alone was detected migrating toward the cathode direction (see the asterisk in Figure 2a). The protein–dsRNA complex was detected at approximately 1200 bp. The mobility shift was dependent on the amount of protein.

To assess any specific binding affinity of TSWV-N protein to DNA, a host insect gene, *Fo-Gn*, in cDNA form was incubated with the TSWV-N protein and the mixture was run on agarose gel (Figure 2b). The cDNA was detected at about 400 bp on the gel, while the complex was shifted to 600 bp. The binding between the protein and cDNA was dependent on the amount of the protein under a constant amount of cDNA. The results indicated that the TSWV-N protein binds to nucleic acids.

To assess the TSWV-N binding affinity to single-stranded RNA (ssRNA), the dsRNA was denatured by heat treatment and used for the mobility shift assay (Figure 2c). As expected, the TSWV-N could bind the denatured dsRNA. Then, we prepared the ssRNA using a single T7 promoter on the cDNA template in either direction (Figure 2d). The directional ssRNA was denoted as + or − and confirmed by their products on the gel. The gel shift assay was applied to assess the binding affinity of TSWV-N to the ssRNA samples (Figure 2e). Both dsRNA and ssRNA were detected at approximately 1.5 kb. In contrast, the protein–RNA complex samples were detected at approximately 3 kb.

### 3.3. TSWV-N Forms a Trimeric Complex to Bind ssRNA

A bioinformatics analysis using AlphaFold v3 indicates that TSWV-N forms a trimeric complex (Figure 3a). The three subunits (protomers A, B, and C) are linked to each other via their N- and C-terminal extensions. In the complex, the core regions allow the binding residues in each protomer to closely contact the ssRNA. To confirm the trimeric conformation, the purified TSWV-N was run on a size-exclusion chromatography column (Figure 3b). Two peaks were separated in earlier fractions (#9–15) and in later fractions (#24–27). When the protein was denatured by heat treatment, the chromatogram resulted in a single later peak. Furthermore, the cross-linking using 0.003% glutaraldehyde resulted in a mostly earlier peak. The SDS-PAGE showed a dimer and trimer along with a monomer protein in the glutaraldehyde-treated sample at the earlier peak (Figure 3c).

### 3.4. Differential Binding of TSWV-N to ssRNA or dsRNA

To compare the binding stability of TSWV-N to two types of RNA samples, ssRNA and dsRNA were incubated with TSWV-N and then treated with RNase A (Figure 4). In the TSWV-N complex with ssRNA, RNase A did not prevent the mobility shift, indicating that TSWV-N protected the ssRNA from the enzyme (Figure 4a). However, the TSWV-N complex with dsRNA was susceptible to RNase A as the gel shift caused by the protein–RNA complex was no longer observed. To clarify the differential susceptibilities, their binding affinities were compared by a thermal shift assay (Figure 4b). With an increase in ambient temperature, TSWV-N lost its folding, which was indicated by the increasing absorbance with the highest peak. The temperature at the maximal peaks of TSWV-N varied depending on whether it was mixed with ssRNA or dsRNA. The temperatures at the maximal peaks were higher in the mixture of TSWV-N and ssRNA compared to the mixture of TSWV-N and dsRNA.

### 3.5. High Affinity of TSWV-N Protein to TSWV Genome

Total RNA was extracted from TSWV-infected or uninfected *F. occidentalis*. Then, the total RNA was incubated with TSWV-N and pulled down using a specific antibody against the recombinant protein (Figure 5a). The RNA samples from the virus-infected thrips exhibited the expression of both viral and insect genes, whereas those from uninfected thrips did not (Figure 5b). When the viral content was compared, the pulled-down sample was relatively rich in the viral RNA compared to the total RNA sample without pull-down with TSWV-N (Figure 5c).

### 3.6. Co-Expression of TSWV Genes in Different Tissues of F. occidentalis

Three viral segments of TSWV were assessed in different tissues of *F. occidentalis* during development using FISH (Figure 6). At the larval stage, all three segments were detected in the midgut and Malpighian tubule (Figure 6a). During the pupal stage, these viral segments were also detected in the salivary gland, along with the midgut and Malpighian tubule. At the adult stage, most viral segments were detected in the salivary gland, with weak expression in the anterior midgut. These expression profiles were quantified using RT-qPCR (Figure 6b). Overall, the three segments displayed similar expression patterns across the developmental stages and tissues.

## 4. Discussion

TSWV-N protein encapsidates the RNA genome to form a ribonucleoprotein complex (RNP). To assess the specificity of TSWV-N to the viral RNA genome, a recombinant TSWV-N protein was prepared using a bacterial expression system and used for binding assays with different nucleic acid molecules. Even though Richmond et al. [22] showed that the protein binds to ssRNA but not dsRNA, our gel shift assays surprisingly showed that the recombinant protein binds to both DNA and RNA. Furthermore, its binding affinity to RNA did not discriminate between dsRNA and ssRNA, regardless of direction. This suggests that the binding of TSWV-N to nucleic acids is not specific but depends on the electrostatic interactions between positively charged protein and negatively charged nucleic acids.

It has been reported that TSWV-N does not discriminate specific sequences of its interacting RNAs [23]. However, our current study suggests a preference of TSWV-N for its viral RNA genome among the total RNAs extracted from TSWV-infected tissues. Compared to the total RNA, the pulled-down RNAs were much richer in viral genomic RNA, although they also contained host insect RNAs. This suggests that TSWV-N has a preference for the viral RNA. However, it remains unclear how the protein discriminates the viral RNA from the total RNA. TSWV-N protein is essential for assembling the viral genomic RNA into the viral RNP, which is involved in various steps of the life cycle of these viruses, including RNA replication, virus particle formation, and cell-to-cell movement during the infection of plant cells by binding affinity to different functional proteins involved in viral transmission and multiplication. For example, TSWV-N is recognized by TSWV-NSm, which mediates the cell-to-cell movement of the nucleoprotein complex [16]. TSWV-N also specifically binds to E3 ubiquitin-protein ligase UBR7, which is closely related to TSWV transmission in *F. occidentalis* [24]. These suggest that TSWV-N exhibits multi-functional bindings by binding to other functional proteins as well as RNA genome.

To further investigate the binding specificity of TSWV-N, its protein–protein interaction was analyzed by size-exclusion chromatography. In its native form the elute was separated into two peaks. However, a chromatogram using the denatured TSWV-N protein showed a single peak at the second elusion volume. Glutaraldehyde was used to cross-link nearby monomers, and the sample was eluted on the chromatography column. This treatment resulted in a single peak at the first elution volume and formed a dimer and trimer in addition to a monomer, as shown by SDS-PAGE. These results suggest that TSWV-N forms multimers. Crystal structure analysis [25] indicates that the TSWV-N protein consists of a main body and two protruding arms at N- (α1 helix domain in Figure 1a) and C- (α13 helix domain in Figure 1a) termini. The main body contains a positively charged groove (red boxes in Figure 1a) as the RNA genome-binding site, while the arms mediate protein–protein interaction to form a ring-shaped homotrimer. The trimer then interacts with another trimer for higher-order oligomerization during the process of virus assembly and packaging [23]. Thus, the dimers and trimers of TSWV-N observed in our current study can be explained by the oligomer property of TSWV-N.

Two different RNPs were compared to further analyze the binding specificity of TSWV-N. The RNP formed from TSWV-N and ssRNA was protected from the RNase A digestion, while the RNP formed from TSWV-N and dsRNA was susceptible to RNase activity. The protective role of TSWV-N against its RNA genome was demonstrated in an earlier study [9]. Our current study showed that TSWV-N failed to protect its RNP containing dsRNA from nuclease activity. To explain this difference, the binding affinity of TSWV-N to the RNA within RNP was assessed using a thermal shift assay. This assay showed that the melting temperature was significantly higher for the ssRNA RNP compared to the dsRNA–RNP complex. These assays suggest that TSWV-N has a higher affinity for ssRNA than for dsRNA, presumably due to tighter oligomerization preferentially on ssRNA.

*TSWV-N* is co-expressed with other viral genes in each tissue of *F. occidentalis* during insect development after viral acquisition. TSWV is transmitted by thrips in a propagative manner [3]. Once the virus infects the larvae through the midgut epithelia, it multiplies and moves to other tissues during host development. Finally, the viral particles reach the salivary gland during the adult stage, exhibiting tissue tropism [26]. Our current study showed the tissue tropism of the virus by demonstrating the multiplication of the three viral genome segments. The co-expression of *TSWV-N* along with other viral segments in different tissues suggests its role in viral multiplication. TSWV-N proteins are localized in the perinuclear region [27], presumably in a RNP form due to their high affinity for the viral RNA genome within a virus-replicating aggregation. Along with its protein–protein interaction and binding affinity for the viral glycoprotein C (G_C_), the protein complex can gain access to the endoplasmic reticulum [28]. Lastly, the binding affinity of G_C_ and glycoprotein N (G_N_) would lead the RNP glycoproteins to enter the Golgi complex to produce the encapsidated viral particles [29]. In addition, the specific terminal sequence of the viral segments may be involved in the specific preference of the capsid protein for the viral RNA genome because the S and M segments contain inverted complementary repeats at their termini, which are probably involved in RNA replication and in the formation of circular nucleocapsids in virions [30].

## 5. Conclusions

This study showed the preferential binding affinity of TSWV-N for the viral genome, consistent with its high affinity for nucleic acids. The co-expression of TSWV-N along with other viral genome segments suggests a virus-producing aggregation in host cells to produce viral particles via protein–protein interactions between TSWV-N and G_C_/G_N_ along with its tissue tropism during sequential developmental stages of *F. occidentalis*. Relative preference of the capsid protein for the viral genome may be reasoned by the presence of the virogenic stroma, which should be explored in future study.

## Figures and Tables

**Figure 1 viruses-17-00826-f001:**
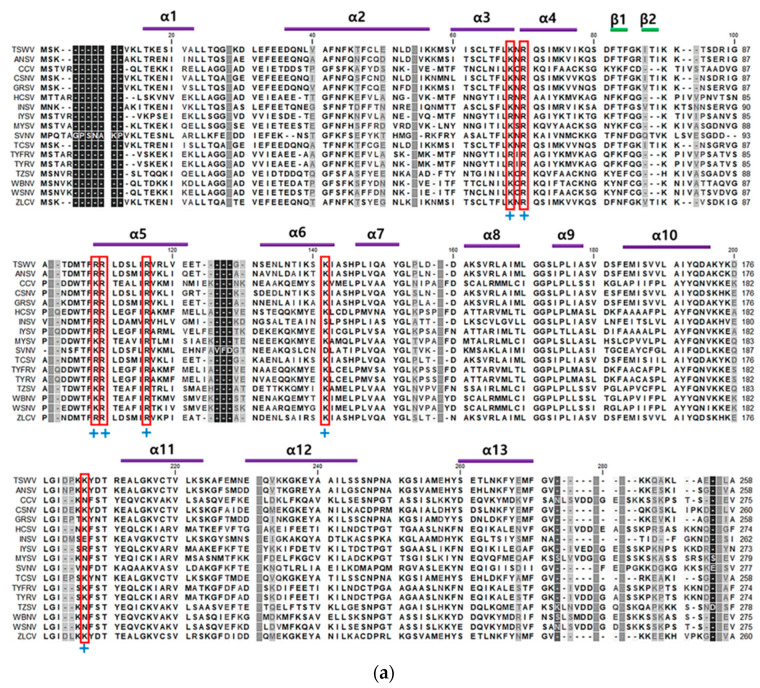
Purification of a recombinant TSWV-N protein using an *E. coli* expression system and affinity chromatography. (**a**) Multiple sequence alignment of the N proteins of 17 orthotospoviruses (Appendix A) using MegAlign with ClustalX program. Conserved amino acid sequences indicated thirteen α-helices and two β-barrel structures. The secondary structures, consisting of 13 alpha helices and 2 beta sheets, are predicted from the TSWV-N protein sequence. DNA/RNA-binding sites are highlighted by a (+) sign. An asterisk indicates the TSWV-N protein migrating toward the anode. (**b**) Purified recombinant TSWV-N protein visualized by Coomassie staining (left panel) and Western blot (right panel). The recombinant protein extract was purified using affinity chromatography with NTA-Ni resin specific to 6× His tag. The insert gene in pBAD-TOPO was overexpressed using 0.2% L-arabinose (L-Ara). Western blot analysis used V6 antibody. An arrow indicates the purified TSWV-N.

**Figure 2 viruses-17-00826-f002:**
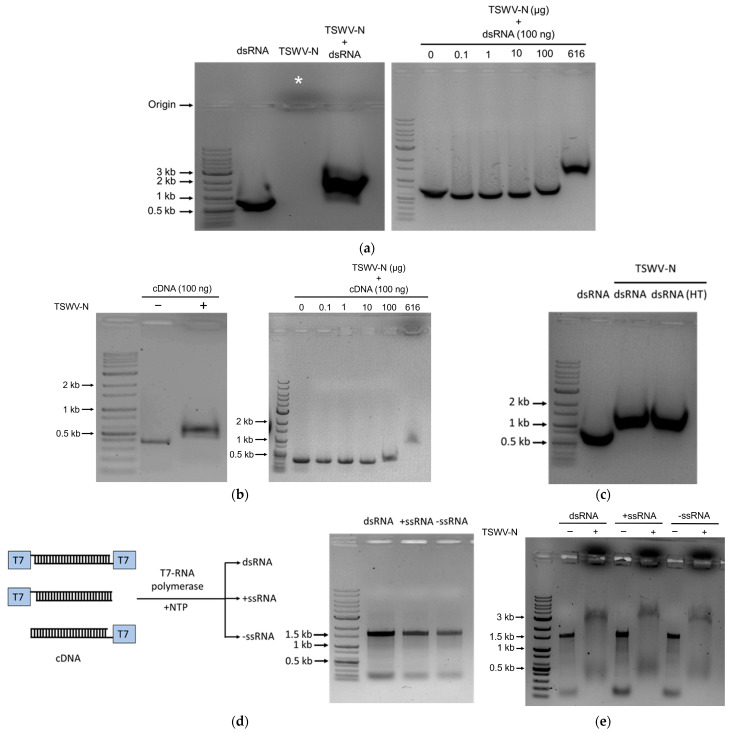
Binding analysis of TSWV-N protein to RNA or DNA using EMSA. A reaction mixture for EMSA consisted of 10 μL of each nucleic acid and TSWV-N protein. The cDNA represented 1.4 kb of *Fo-Gn* cDNA. RNAs were prepared from the *Fo-Gn* cDNA template. (**a**) Binding affinity for dsRNA. Left panel: dsRNA (1 µg), TSWV-N protein (6.16 µg), or their mixture. Right panel: different amounts of TSWV-N protein and dsRNA (1 µg). (**b**) Binding affinity for cDNA. Left panel: cDNA (1 µg), TSWV-N protein (6.16 µg), or their mixture. Right panel: different amounts of TSWV-N and cDNA (1 µg). (**c**) Binding affinity for denatured dsRNA. The dsRNA sample was treated with heat (‘HT’, 95 °C for 5 min). dsRNA or heated dsRNA (1 µg), TSWV-N protein (6.16 µg), or their mixtures. (**d**) Preparation of single-stranded RNAs derived from 1.4 kb of *Fo-Gn* cDNA template using T7 RNA polymerase: T7 tags at both 5′ ends for dsRNA production and a T7 tag at a single 5′ end for positive or negative (+/−) ssRNA production. RNA production was confirmed by 1% agarose gel electrophoresis. (**e**) Binding affinity for ssRNA. dsRNA (1 µg) alone (−) or its mixture (+) with TSWV-N protein (6.16 µg). All EMSA mixtures were incubated for 1 min at room temperature and separated on 1% agarose gel.

**Figure 3 viruses-17-00826-f003:**
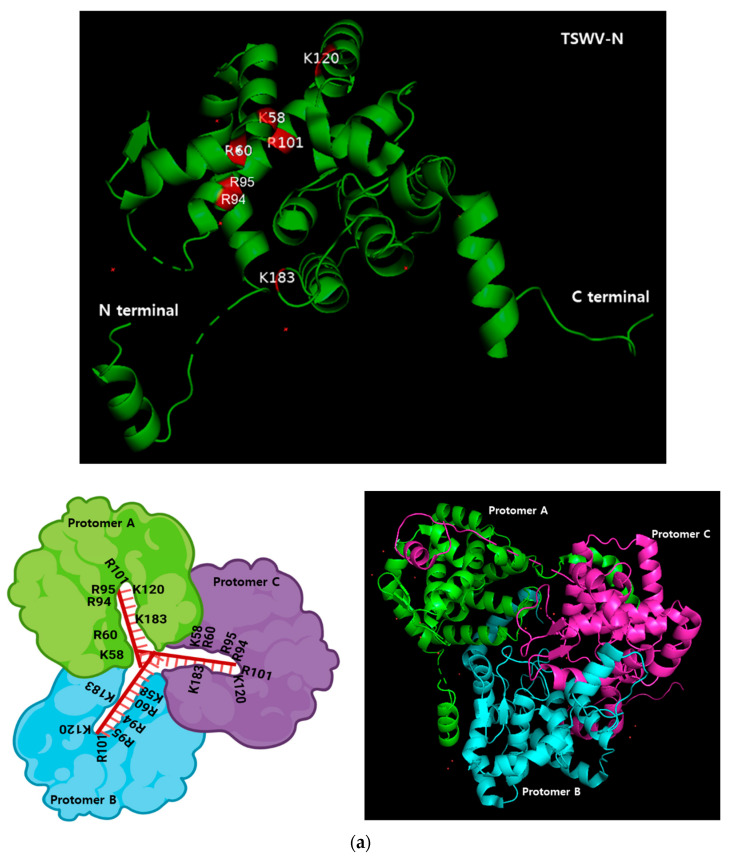
Oligomer formation of the recombinant TSWV-N protein. (**a**) Prediction of its oligomerization. A 3D structure of the recombinant protein was predicted using AlphaFold v3 (https://neurosnap.ai/, accessed on 15 May 2025), showing its trimeric structure. The RNA-binding sites (seven residues) predicted in Figure 1a are located on the inside of the protein. The three polypeptides are labeled as protomer A, B, and C. A schematic diagram was drawn using Biorender to illustrate the DNA/RNA binding site in the TSWV-N protein. (**b**) Size-exclusion chromatography to separate different oligomers of TSWV-N using a Sephadex G-100 column. Denatured TSWV-N (‘HT’) protein was prepared by heat treatment at 95 °C for 5 min. Cross-linking of oligomers was performed with 0.003% glutaraldehyde (‘GLA’) at 37 °C for 5 min. (**c**) Oligomeric structures of the early peak separated on 10% SDS-PAGE.

**Figure 4 viruses-17-00826-f004:**
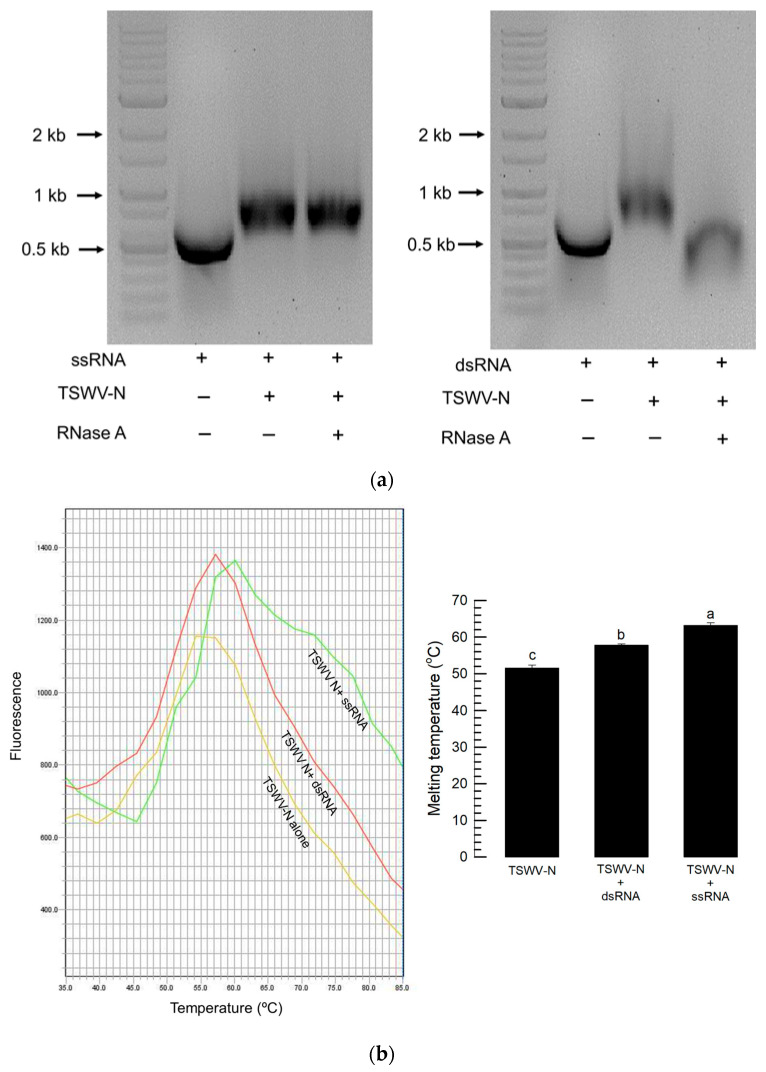
Differential binding affinities of the recombinant TSWV-N protein to ssRNA and dsRNA. (**a**) Differential susceptibilities of TSWV-N mixtures with ssRNA or dsRNA to RNase A digestion. After 1 min digestion, the mixtures were run on 1% agarose gel. (**b**) Binding affinity assessment of TSWV-N and RNA by thermal shift assay. The melting curves were obtained for TSWV-N alone and its mixtures with dsRNA or ssRNA. A melting curve experiment was conducted with these mixtures using an increasing temperature gradient at a rate of 2.9 °C/min from 25 °C to 95 °C. The melting temperature was determined from the temperature at maximal fluorescence. Each treatment was replicated three times. Different letters following the standard deviation indicate significant differences among means at a Type I error = 0.05 (LSD test).

**Figure 5 viruses-17-00826-f005:**
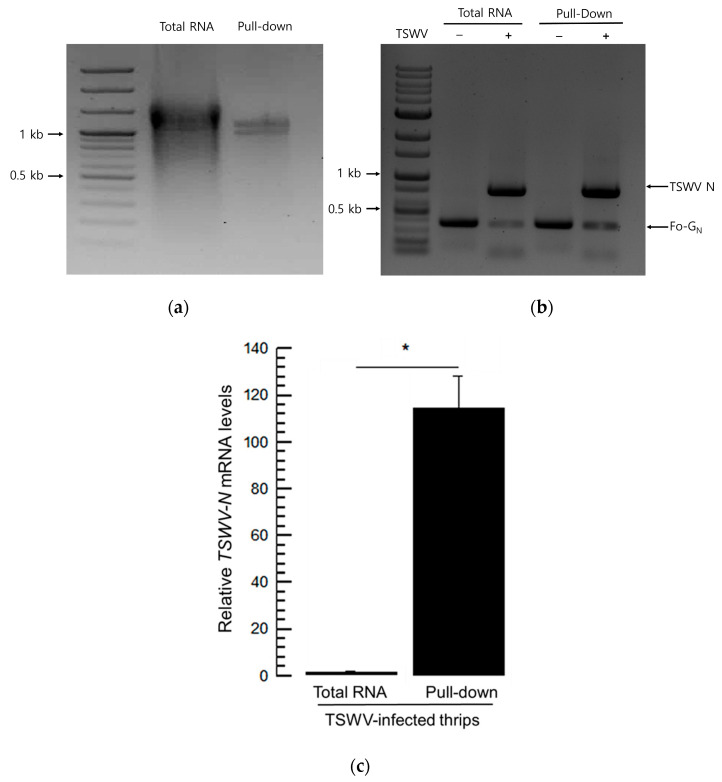
Preference of the recombinant TSWV-N for the viral genome. (**a**) Extraction of RNAs binding to TSWV-N by pull-down assay. Total RNA was extracted from F. occidentalis that was infected with TSWV and incubated with TSWV-N protein. The RNA–protein mixture was pulled-down with Ni-NTA resin and subjected to RNA extraction. (**b**) Multiplex PCR with primers specific to Fo-Gn and TSWV-N genes. ‘−’ represents RNA extracts from *F. occidentalis* uninfected with TSWV. ‘+’ represents RNA extract from the infected thrips. (**c**) Comparison of the relative TSWV genome titers between total RNA extract and TSWV-N-assisted pull-down RNA extract from the viruliferous thrips. The quantification used RT-qPCR with a target gene, TSWV-N. An asterisk (*) indicates significant differences between two treatments at a Type I error = 0.05 (LSD test).

**Figure 6 viruses-17-00826-f006:**
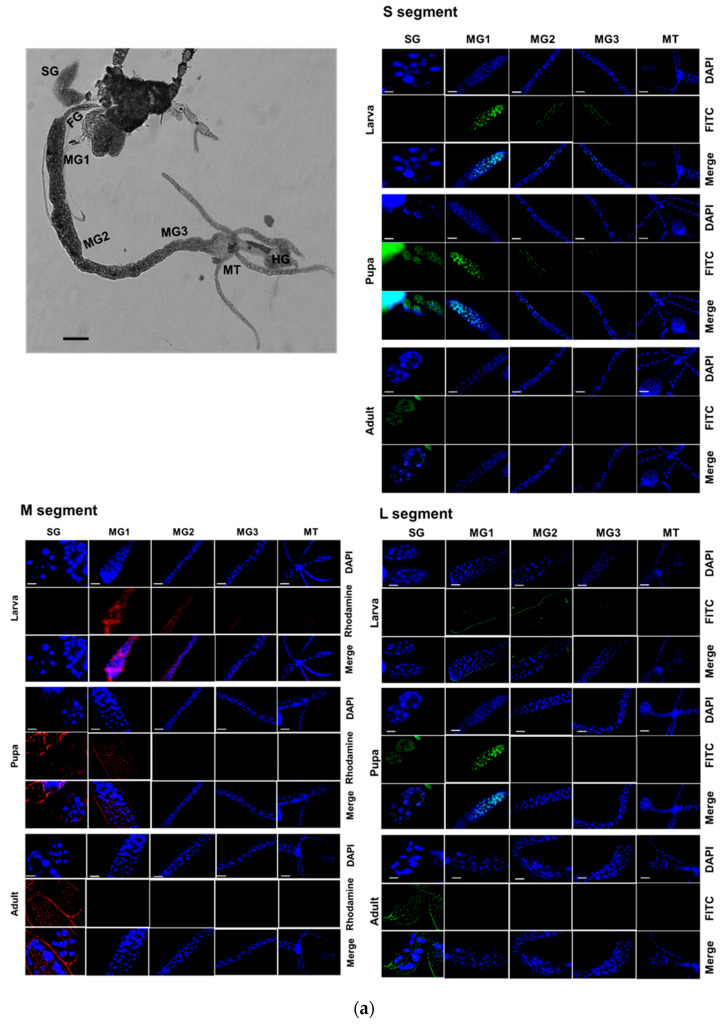
Replication of TSWV genomes in different tissues at three developmental stages of F. occidentalis. (**a**) Fluorescence *in situ* hybridization (FISH) analysis of three viral segments large (‘L’), medium (‘M’), and small (‘S’). Tissues include the foregut (‘FG’), midgut (‘MG1′, ‘MG2′, and ‘MG3′), Malpighian tubules (‘MT’), hindgut (‘HG’), and salivary gland (‘SG’). Scale bar represents 0.1 mm. FISH was conducted using fluorescein amidites (FAM)-labeled probes (green-colored) specific to S segment with TSWV-N and L segment with RdRp gene. A rhodamine 6G-labeled probe (red-colored) was used to detect the M segment with NSm gene. Nucleus was stained with DAPI (blue-colored) (**b**) Relative intensities of the three viral segments in different tissues at three developmental stages. Different letters above the standard deviation bars indicate significant differences among means at Type I error rate of 0.05 (LSD test).

## Data Availability

The original contributions presented in the study are included in the article/Appendix A, further inquiries can be directed to the corresponding author.

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
