# Peer review of "Binding Specificity and Oligomerization of TSWV N Protein in the Western Flower Thrips, Frankliniella occidentalis"

_viruses, 2025, doi:10.3390/v17060826_

Round 1

Reviewer 1 Report

Comments and Suggestions for Authors

This manuscript addresses the molecular interactions between Tomato spotted wilt virus (TSWV) nucleocapsid (N) protein and various forms of nucleic acids, with a focus on its binding specificity and oligomerization in the insect vector Frankliniella occidentalis. The study integrates biochemical assays, in silico modeling, and FISH-based localization, providing a multifaceted view of the N protein’s role in viral replication and tropism.

The topic is timely and scientifically relevant, but the manuscript requires substantial revisions to improve clarity, scientific rigor, and structural organization. Below are major and minor issues that should be addressed before the paper can be considered for publication.

  1. EMSA and pull-down assays lack rigorous controls. It is unclear whether the observed “preference” for viral RNA is due to sequence specificity or relative abundance. Please include binding assays using total RNA from non-infected plants/insects or unrelated viral RNAs to test specificity. Also, normalize RNA input amounts.
  2.  Some figures (e.g., Figure 2 and 5) lack quantification of band intensities. This weakens conclusions about relative binding affinities. Please provide densitometric analysis of gel bands and include statistical comparisons for at least three biological replicates.
  3. The conclusion that ssRNA binds more strongly than dsRNA based on thermal shift is overstated. The melting temperature changes are not clearly linked to specific binding without proper binding curves or KD estimation. Please include control curves (protein alone, RNA alone), explain melting curve interpretation more clearly, and consider a binding affinity assay with quantifiable output (e.g., SPR or ITC).
  4. The Results and Discussion sections have redundancies and logical jumps. Some results are described without proper figure references. Please streamline the Results section to avoid repetition and improve logical flow (e.g., binding assays → oligomerization → tissue localization). Link each experimental result directly to a figure.
  5. FISH data are interesting but insufficiently explained. There is no quantification of signal intensity or statistical validation of localization differences. Please provide a clearer quantification of FISH signal (e.g., using ImageJ), add scale bars, and describe sample size and tissue sectioning method.
  6. The current title is too long. Consider shortening it to better reflect the main findings. “Binding specificity and oligomerization of Tomato spotted wilt virus N protein in Frankliniella occidentalis.”
  7. Some gel images (e.g., Figures 2 and 4) are low-resolution, and arrows or labels are missing. Please  improve image quality and ensure all lanes are clearly annotated.
  8. Please provide catalog numbers and vendors for key reagents (e.g., T7 RNA polymerase, antibodies). 
  9. Many references are outdated or redundant. Incorporate more recent literature, especially studies applying structural or biophysical analyses of TSWV proteins. Please replace or supplement with studies published after 2020.

Author Response

Comment #1-1: This manuscript addresses the molecular interactions between Tomato spotted wilt virus (TSWV) nucleocapsid (N) protein and various forms of nucleic acids, with a focus on its binding specificity and oligomerization in the insect vector Frankliniella occidentalis. The study integrates biochemical assays, in silico modeling, and FISH-based localization, providing a multifaceted view of the N protein’s role in viral replication and tropism. The topic is timely and scientifically relevant, but the manuscript requires substantial revisions to improve clarity, scientific rigor, and structural organization. Below are major and minor issues that should be addressed before the paper can be considered for publication.

Response: Based on your comments, all issues are carefully addressed.

Comment #1-2: EMSA and pull-down assays lack rigorous controls. It is unclear whether the observed “preference” for viral RNA is due to sequence specificity or relative abundance. Please include binding assays using total RNA from non-infected plants/insects or unrelated viral RNAs to test specificity. Also, normalize RNA input amounts.

Response: It is a nice comment. The results showed that there is little specificity of the binding affinity of N protein and nucleic acids. We used both insect DNA/RNA (Fig. 2a-c) and virus DNA/RNA (Fig. 2d-e) for the binding assays. For normalization, all EMSA used the same amount of protein and nucleic acids except quantitative binding assays using different ligand amounts.

Comment #1-3: Some figures (e.g., Figure 2 and 5) lack quantification of band intensities. This weakens conclusions about relative binding affinities. Please provide densitometric analysis of gel bands and include statistical comparisons for at least three biological replicates.

Response: In Figure 2, this EMSA assays were designed to see the band shifting by binding affinity of N protein in a constant nucleic acid amount. In all assays, the bands were shifted by binding. Figure 5A shows the subsequent extraction of total RNAs. Fig. 5B shows the viral and insect RNAs by RT-PCR. Fig. 5C shows RT-qPCR by normalization of internal gene expression.

Comment #1-4: The conclusion that ssRNA binds more strongly than dsRNA based on thermal shift is overstated. The melting temperature changes are not clearly linked to specific binding without proper binding curves or KD estimation. Please include control curves (protein alone, RNA alone), explain melting curve interpretation more clearly, and consider a binding affinity assay with quantifiable output (e.g., SPR or ITC).

Response: The additional data about the controls are newly provided in Supplementary Information about the thermal shift assays using RNA alone. In addition, the figure is revised by replacing ‘No TSWV’ with ‘TSWV-N alone’.

Comment #1-5: The Results and Discussion sections have redundancies and logical jumps. Some results are described without proper figure references. Please streamline the Results section to avoid repetition and improve logical flow (e.g., binding assays → oligomerization → tissue localization). Link each experimental result directly to a figure.

Response: It is a nice suggestion. Fourth paragraph is logically related with the first paragraph. Thus these are relocated.

Comment #1-6: FISH data are interesting but insufficiently explained. There is no quantification of signal intensity or statistical validation of localization differences. Please provide a clearer quantification of FISH signal (e.g., using ImageJ), add scale bars, and describe sample size and tissue sectioning method.

Response: As you see, Fig 6a indicates images and Fig 6b explains the FISH intensities.

Comment #1-7: The current title is too long. Consider shortening it to better reflect the main findings. “Binding specificity and oligomerization of Tomato spotted wilt virus N protein in Frankliniella occidentalis.”

Response: It is a nice comment. As per comment, we have changed the title as follow: “Binding specificity and oligomerization of TSWV N protein in the western flower thrips, Frankliniella occidentalis”

Comment #1-8: Some gel images (e.g., Figures 2 and 4) are low-resolution, and arrows or labels are missing. Please improve image quality and ensure all lanes are clearly annotated.

Please provide catalog numbers and vendors for key reagents (e.g., T7 RNA polymerase, antibodies).

Response: We confirmed the arrows. We also confirmed the reagents at the first by adding address.

Comment #1-9: Many references are outdated or redundant. Incorporate more recent literature, especially studies applying structural or biophysical analyses of TSWV proteins. Please replace or supplement with studies published after 2020.

Response: Two (16, 29) references are updated.

Reviewer 2 Report

Comments and Suggestions for Authors

The paper reports the following:

(1) an analysis of the RNA-binding properties of the TSWV N protein, and

(2) the distribution of the three TSWV genomic RNAs in the thrips, the virus vector.

In my opinion, the paper cannot be considered for publication in Viruses in its current form and requires significant revisions. The main points of criticism are listed below.

  1. The English is unacceptable. Nearly every sentence uses inappropriate wording. The entire manuscript should be rewritten, preferably with the help of a native speaker who is competent in this field or, at the very least, in the field of molecular biology.

  1. Section 3.6: The title and first sentence are incorrect and misleading. The detection of TSWV RNAs in the insect vector is unrelated to the monitoring of TSWV-N expression that the authors announced. In fact, such monitoring was not performed. Therefore, the second part of the title, "its co-expression with other TSWV genes along with its tissue tropism," refers to results not presented in the paper. Thus, the title should be changed to align with the actual results reported in the paper. Additionally, the title should be shortened. Moreover, the content of this section is not related to the main subject of the paper, which is the binding of RNA by the N protein.

  1. There are concerns regarding the RNA substrates used for gel shift assays.

a) It should be clarified why the thrips gene MH884757.1 was used.

b) Why was this RNA used in double-stranded form when it is a protein-coding mRNA that never exists as dsRNA in cells? This should be explained in the paper.

c) The N (nucleocapsid) protein binds viral RNA; therefore, it is unclear why binding to a vector RNA was tested while binding to the natural substrate was not. Binding of viral RNA should be presented in the paper.

d) Why did the authors not use a negative control? (e.g., GFP transcript).

Additional experiments are required to resolve these issues.

  1. It is not clear why the authors state that the N protein binds RNA in a trimeric form. Indeed, it exists as trimers in solution, this does not mean however that RNA is bound by trimers. Based on the information on other N proteins, the trimers can be reorganized into multimers upon RNA binding.

  1. Figures should be reworked to be more compact. Figure 2 is most unacceptable in its layout. In other figures, the individual panels can easily be made smaller while their positions can be arranged more thoughtfully.

Author Response

Comment #2-1: The paper reports the following:

(1) an analysis of the RNA-binding properties of the TSWV N protein, and

(2) the distribution of the three TSWV genomic RNAs in the thrips, the virus vector.

In my opinion, the paper cannot be considered for publication in Viruses in its current form and requires significant revisions. The main points of criticism are listed below.

Response: Based on your comments, all issues are carefully addressed.

Comment #2-2: The English is unacceptable. Nearly every sentence uses inappropriate wording. The entire manuscript should be rewritten, preferably with the help of a native speaker who is competent in this field or, at the very least, in the field of molecular biology.

Response: The whole text was revised by the authors after revision to keep a context consistency.

Comment #2-3: Section 3.6: The title and first sentence are incorrect and misleading. The detection of TSWV RNAs in the insect vector is unrelated to the monitoring of TSWV-N expression that the authors announced. In fact, such monitoring was not performed. Therefore, the second part of the title, "its co-expression with other TSWV genes along with its tissue tropism," refers to results not presented in the paper. Thus, the title should be changed to align with the actual results reported in the paper. Additionally, the title should be shortened. Moreover, the content of this section is not related to the main subject of the paper, which is the binding of RNA by the N protein.

Response: (1) 3.6 subtitle is changed as follows: “Co-expression of TSWV genes in different tissues of F. occidentalis”

(2) The first sentence is rephrased as follows: “Three viral segments of TSWV were assessed in different tissues of F. occidentalis during development using FISH (Figure 6).”

(3) Title is changed into as follows: “Binding specificity and oligomerization of TSWV N protein in the western flower thrips, Frankliniella occidentalis”

Comment #2-4: There are concerns regarding the RNA substrates used for gel shift assays.

  1. a) It should be clarified why the thrips gene MH884757.1 was used.

Response: To clarify the specificity, we used viral and insect genes in RNA and DNA forms. The sentence is rephrased as follows: “To assess any specific binding affinity of TSWV-N protein to DNA, a host insect gene, Fo-Gn, in a cDNA form was incubated with the TSWV-N protein and run on the agarose gel (Figure 2b).”

  1. b) Why was this RNA used in double-stranded form when it is a protein-coding mRNA that never exists as dsRNA in cells? This should be explained in the paper.

Response: As you see Fig. 2d-2e, the binding assays used ssRNA or dsRNA.

  1. c) The N (nucleocapsid) protein binds viral RNA; therefore, it is unclear why binding to a vector RNA was tested while binding to the natural substrate was not. Binding of viral RNA should be presented in the paper.

Response: It is a nice comment. Unexpectedly, our results showed that the nucleocapsid protein can bind to any forms of nucleic acids. However, the protein has preference for the viral RNA. This preference was not addressed in this study. This is described in Discussion.

  1. d) Why did the authors not use a negative control? (e.g., GFP transcript).

Additional experiments are required to resolve these issues.

Response: We used both viral and insect nucleic acids. As mentioned, the protein binds to both insect and viral DNA/RNA though it should bind only to viral RNA. Thus, the insect DNA/RNA were the controls in this study.

Comment #2-5: It is not clear why the authors state that the N protein binds RNA in a trimeric form. Indeed, it exists as trimers in solution, this does not mean however that RNA is bound by trimers. Based on the information on other N proteins, the trimers can be reorganized into multimers upon RNA binding.

Response: Our bioinformatics suggests that TSWV-N forms trimeric complex. Indeed, the recombinant protein in this forms multimers in native form from our gel filtration chromatography. However, it remains that the protein may form trimer with RNA genome in the virus. It should be explored in future study.

Comment #2-6: Figures should be reworked to be more compact. Figure 2 is most unacceptable in its layout. In other figures, the individual panels can easily be made smaller while their positions can be arranged more thoughtfully.

Response: Fig 2 is reduced in size to be in a page.

Reviewer 3 Report

Comments and Suggestions for Authors

Authors Falguni Khan and coworkers presented here a manuscript entitled „Binding affinity of the capsid N protein to different nucleic acids and its co-expression with other TSWV genes along with its tissue tropism in the western flower thrips, Frankliniella occidentalis“.

The main contribution of the manuscript is the analysis of the properties of the N protein of tomato spotted wilt virus. The authors performed a series of analyses and elucidated previously unknown functions of the N protein in relation to nucleic acids in general and specifically.

The paper is well written and the methods and procedures used are well described.

The results are presented in great detail.

The discussion briefly summarizes the results and compares them with previous work.

The authors have supplemented their work with a number of figures and graphs that illustrate the results obtained.

I recommend writing a brief conclusion as a new subsection.

I have no other comments on the paper.

Author Response

Comment: Authors Falguni Khan and coworkers presented here a manuscript entitled „Binding affinity of the capsid N protein to different nucleic acids and its co-expression with other TSWV genes along with its tissue tropism in the western flower thrips, Frankliniella occidentalis“. The main contribution of the manuscript is the analysis of the properties of the N protein of tomato spotted wilt virus. The authors performed a series of analyses and elucidated previously unknown functions of the N protein in relation to nucleic acids in general and specifically. The paper is well written and the methods and procedures used are well described. The results are presented in great detail. The discussion briefly summarizes the results and compares them with previous work. The authors have supplemented their work with a number of figures and graphs that illustrate the results obtained. I recommend writing a brief conclusion as a new subsection. I have no other comments on the paper.

Response: We appreciate the thoughtful comments to upgrade our manuscript. We add a new section for conclusion according to the comment.

Round 2

Reviewer 1 Report

Comments and Suggestions for Authors

The overall quality and clarity of the figures have improved in the revised manuscript. However, I would like to point out a potential issue in Figure 3, specifically the western blot image in panel (c). The bands appear somewhat saturated, which may obscure subtle differences in signal intensity between oligomeric forms.

While the manuscript presents a well-structured experimental framework and diverse data, the Discussion and Conclusion sections appear relatively brief considering the scope and significance of the study. The Discussion could be strengthened by incorporating deeper comparisons with previous studies and elaborating on the biological or mechanistic implications of the findings. The Conclusion would also benefit from additional content that addresses the broader impact of the study, potential applications (e.g., virus-vector interaction control), and suggestions for future research directions.

Author Response

Comment #1-1: The overall quality and clarity of the figures have improved in the revised manuscript. However, I would like to point out a potential issue in Figure 3, specifically the western blot image in panel (c). The bands appear somewhat saturated, which may obscure subtle differences in signal intensity between oligomeric forms.

Response: Fig. 3C is not a western blot but a Coomassie staining of the oligomers cross-linked by glutaraldehyde treatment. The difference in the band intensity represents the variation in the oligomeric forms.

Comment #1-2: While the manuscript presents a well-structured experimental framework and diverse data, the Discussion and Conclusion sections appear relatively brief considering the scope and significance of the study. The Discussion could be strengthened by incorporating deeper comparisons with previous studies and elaborating on the biological or mechanistic implications of the findings. The Conclusion would also benefit from additional content that addresses the broader impact of the study, potential applications (e.g., virus-vector interaction control), and suggestions for future research directions.

Response:

(1) Additional Discussion: We add the binding specificity of TSWV-N as follows: “In addition, the specific terminal sequence of the viral segments may be involved in the specific preference of the capsid protein for the viral RNA genome because the S and M segments contain inverted complementary repeats at their termini, probably involved in RNA replication and in the formation of circular nucleocapsids in virions [30].”

(2) Additional Conclusion: We add a future research direction based on the current study as follows: “Relative preference of the capsid protein for the viral genome may be reasoned by the presence of the virogenic stroma, which should be explored in future study.”

Reviewer 2 Report

Comments and Suggestions for Authors

The paper can be accepted for publication in its current form.

Author Response

Comment #2-1: The paper can be accepted for publication in its current form.

Response: Thanks